# Effect of Aucubin-Containing Eye Drops on Tear Hyposecretion and Lacrimal Gland Damage Induced by Urban Particulate Matter in Rats

**DOI:** 10.3390/molecules27092926

**Published:** 2022-05-04

**Authors:** Su-Bin Park, Woo Kwon Jung, Hwa-Young Yu, Yong Hwan Kim, Junghyun Kim

**Affiliations:** Department of Oral Pathology, School of Dentistry, Jeonbuk National University, Jeonju 54896, Korea; tnqls309@gmail.com (S.-B.P.); wkjungjbnu@gmail.com (W.K.J.); naive17jbnu@gmail.com (H.-Y.Y.); kg8229ku@gmail.com (Y.H.K.)

**Keywords:** aucubin, corneal irregularity, dry eye, lacrimal gland, particulate matter, tear secretion

## Abstract

Exposure to particulate matter is a causative factor of dry eye disease. We aimed to investigate the beneficial effect of eye drops containing aucubin on dry eye disease induced by urban particulate matter (UPM). Dry eye was induced in male SD rats (6 weeks old) by topical exposure to UPM thrice a day for 5 d. Eye drops containing 0.1% aucubin or 0.5% aucubin were topically administered directly into the eye after UPM exposure for an additional 5 d. Tear secretion was evaluated using a phenol red thread tear test and corneal irregularity. The oxidative damage in the lacrimal gland was evaluated using TUNEL and immunohistochemical staining. The topical administration of aucubin significantly attenuated UPM-induced tear hyposecretion (control group: 9.25 ± 0.62 mm, UPM group: 4.55 ± 0.25 mm, 0.1% aucubin: 7.12 ± 0.58 mm, and 0.5% aucubin: 7.88 ± 0.75 mm) and corneal irregularity (control group: 0.00 ± 0.00, UPM group: 3.40 ± 0.29, 0.1% aucubin: 1.80 ± 0.27, and 0.5% aucubin: 1.15 ± 0.27). In addition, aucubin also reduced the UPM-induced apoptotic injury of lacrimal gland cells induced by oxidative stress through the increased expression of HMGB1 and RAGE. These findings indicate that the topical administration of aucubin eye drops showed a beneficial effect against UPM-induced abnormal ocular changes, such as tear hyposecretion and lacrimal gland damage. Therefore, our results reveal the pharmacological activities of aucubin in dry eye disease.

## 1. Introduction

Dry eye disease is one of the most common ophthalmic diseases caused by a disturbance of the functional unit of tear secretion consisting of the ocular surface (cornea, conjunctiva, and conjunctival blood vessels), lacrimal glands, and interconnecting innervations [1,2]. More than 30% of the global population across various ages has dry eye syndrome [1], and dry eye patients suffer from poor quality of life [3]. Artificial tears are the first therapeutic option for dry eye disease but only partially replenish the aqueous tear fluid [4]. Thus, it is necessary to administer multiple topical applications.

Particulate matter is very small solid or liquid particles present in the surrounding air [5]. Recently, particulate matter has become well known as an inducer of environmental pollution and problems or diseases of the skin and eyes [6]. Furthermore, urban particulate matter (UPM) has a harmful effect that causes oxidative stress and inflammation [6]. While many studies have associated UPM with respiratory diseases [7,8,9], few previous studies have focused on ocular diseases such as dry eye disease. In addition, the mechanisms linking UPM and dry eye disease have not been fully elucidated.

Some natural and synthetic compounds have beneficial effects on dry eye disease [10]. *Aucuba japonica* Thunb. is a traditional medicinal herb known for its effects on inflammation and edema [11]. Aucubin is a component isolated from *A. japonica* and is iridoid glucose (Figure 1A). Aucubin is a phytocompound with potent antioxidant, anti-inflammatory, anti-microbial, anti-analgesic, and anti-tumor effects [12,13,14]. We previously reported that the oral administration of aucubin protected against dry eye disease induced by unilateral lacrimal gland excision in rats [15]. However, oral administration of some phytocompounds was absorbed through the small intestine and spread throughout the body, particularly in the ocular tissues [16,17]. However, the absorption of phytocompounds through the digestive tract is often hindered by various components, such as nutrients, ions, fiber, other phytocompounds, and even drugs [18]. The topical administration of eye drops containing catechins, ferulic acid, or several traditional herbal extracts is effective in humans and animal models [19]. In this study, we investigated whether the topical administration of eye drops containing aucubin is effective against UPM-induced dry eye disease, including changes in tear volume, corneal irregularity, and lacrimal gland damage.

## 2. Results

### 2.1. Aucubin Improves Hyposecretion of Tear in Rats with UPM-Induced Dry Eye

The changes in tear secretion caused by aucubin eye drops were evaluated in UPM-induced dry eye rats. As shown in Figure 1A, tear volume decreased in the UPM group (4.55 ± 0.25 mm) compared to that of the control group (9.25 ± 0.62 mm). This reduction in tear secretion by UPM was dose-dependently reversed via the topical administration of 0.1% aucubin (7.12 ± 0.58 mm) and 0.5% aucubin (7.88 ± 0.75 mm). These data suggest that UPM-treated rats had lacrimal hypofunction due to decreased tear secretion, which can be alleviated by aucubin. To evaluate the tear film stability, the corneal irregularity score was measured. As shown in Figure 1B, the circular shape of the ring illuminator was clearly maintained on the corneal surface in the control group (0.00 ± 0.00), but this circular shape was severely distorted in the UPM group (3.40 ± 0.29). The topical administration of aucubin eye drops inhibited this distortion of the circular shape of the ring illuminator (1.80 ± 0.27 and 1.15 ± 0.27, respectively).

### 2.2. Aucubin Reduces Oxidative Injury in the Lacrimal Gland

To identify the oxidative injury in the lacrimal glands of UPM-treated rats, we performed the ROS assay and immunohistochemistry of 8-OHdG, a marker for oxidative DNA damage, in the lacrimal gland. As shown in Figure 2, ROS production and 8-OHdG expression were highly increased in the UPM-treated rats. The topical administration of aucubin eye drops dose-dependently inhibited the generation of ROS and oxidative DNA damage in the UPM-treated rats.

### 2.3. Aucubin Prevents Apoptotic Injury in the Lacrimal Glands of UPM-Treated Rats

Several previous studies have shown that lacrimal cells were damaged by apoptotic injury under dry eye conditions [20,21]. Thus, TUNEL staining was performed to confirm that aucubin eye drops inhibit the apoptosis of lacrimal cells. As shown in Figure 3, the apoptosis of lacrimal cells was increased in the UPM group and restored by aucubin eye drops in a dose-dependent manner. Aucubin protects lacrimal gland tissues by reducing apoptosis.

### 2.4. Aucubin Inhibits the Expression of HMGB1 in the Lacrimal Gland in UPM-Treated Rats

HMGB1 acts as an alarm signal for tissue injury [22]. To confirm the oxidative-stress-induced lacrimal gland damage, the expression of HMGB1 was evaluated. In the control groups, HMGB1 was expressed at low levels only in the nuclei of acinar cells. However, HMGB1 was highly detected in the nuclei and cytoplasm of acinar cells in the UPM group. However, the increased HMGB1 expression in the UPM-treated rats was significantly reduced by the treatment of aucubin eye drops (Figure 4A,B). The expression of HMGB1 mRNA was also evaluated to confirm the experimental results. The mRNA expression levels of HMGB1 were significantly decreased by the treatment of aucubin eye drops in the UPM-treated rats (Figure 4C).

### 2.5. Aucubin Decreases RAGE Expression in the Lacrimal Gland in UPM-Treated Rats

RAGE is a multiligand receptor that binds structurally diverse molecules, including HMGB1 [23] View Record in Scopus|Cited By in Scopus (441). The expression levels of RAGE were evaluated in the lacrimal glands using immunohistochemistry. The RAGE expression level was highly expressed in the UPM-treated rats’ lacrimal glands compared to those of the control rats. This increase in the RAGE protein’s positive signal was significantly decreased by the treatment of aucubin eye drops (Figure 5A,B). The mRNA expression levels of RAGE were also significantly decreased by the treatment of aucubin eye drops in the UPM-treated rats (Figure 5C).

## 3. Discussion

The increase in particulate matter is becoming a serious environmental problem worldwide. Particle matter is classified by size as follows: ultrafine, fine, and coarse particles [24]. The maximum diameter of ultrafine particulate matter is 0.1 µm (PM0.1), fine particulate matter is 2.5 µm (PM2.5), and coarse particulate matter is 10 µm (PM10). Among them, coarse UPM (SRM 1648a) has been used to evaluate the harmful role of particulate matter [25,26,27,28]. Based on this existing research, we focused on UPM-related dry eye disease.

Since the eyes are external organs that can be easily exposed to particulate matter, they are susceptible to ocular surface disorders such as dry eye [1]. Dry eye disease is caused by a dysfunction of the lacrimal gland. Dysfunction of the lacrimal gland elicits the impairment of the tear film and injury of the cornea and conjunctiva [29]. Moreover, damage to the corneal epithelium and conjunctival epithelium causes dysfunction of the ocular barrier, which eventually worsens dry eye syndrome [29].

In several previous studies, the topical exposure of PM2.5 on the ocular surface of mice induced the reduction of tear secretion, and it also induced dry-eye-related, abnormal changes on the ocular surface [30]. Song et al. reported that the aerosol exposure of urban PM to the ocular surface of rats affected the tear film stability and caused a decrease in tear volume [31]. When PM was exposed to the ocular surface of normal and dry eye rats, the ocular surface of rats with dry eyes had a more severe injury compared to those with normal eyes [32]. Mu et al. also reported that the aerosol exposure of PM to rats caused the injury of corneal cells and conjunctiva cells [33]. Topical exposure to PM2.5 resulted in inflammation and cellular injury in the lacrimal gland, cornea, and conjunctiva [34]. These abnormal changes in the eyes of an experimental animal exposed to PM are similar to the clinical signs of patients with dry eyes. The topical exposure of PM2.5 to the ocular surface of rats has been used as an animal model for dry eye disease [33]. In our study, this PM2.5-induced, dry eye rat was used to evaluate the efficacy of aucubin on the hypofunction of the lacrimal gland.

Aucubin has several pharmacological activities, such as antioxidant, anti-inflammatory, anti-microbial, anti-analgesic, and anti-tumor effects [12,13,14]. In our previous report, the oral administration of aucubin has a beneficial effect on dry eye disease induced by unilateral lacrimal gland excision in rats [15]. In the present study, the topical administration of aucubin eye drops showed a beneficial effect against UPM-induced, abnormal ocular changes, such as tear hyposecretion and lacrimal gland damage. Therefore, our results reveal the pharmacological activities of aucubin in dry eye disease.

The ocular surface is always covered with a thin tear film [35], and the stability of this tear film plays a very important role in maintaining the constancy of the ocular surface [29]. The lacrimal gland secretes a large fraction of the aqueous tear fluid. The tear film consists of three main layers. The inside is the mucin layer, the middle is the aqueous layer, and the outside is the lipid layer. An insufficient aqueous layer, induced by lacrimal gland dysfunction, is a prominent cause of dry eye [36]. Therefore, the pharmacological treatment strategy for dry eye disease is focused on restoring the tear secretion and lacrimal gland function [36]. In this respect, our results suggest that aucubin eye drops have a beneficial effect on tear secretion through the prevention of lacrimal gland damage in PM2.5-exposed eyes.

The overproduction of ROS causes inflammation and apoptosis through various signaling pathways in our body [37,38,39]. Oxidative, stress-related cellular damage was observed in the lacrimal glands of Sod1^−/−^ mice [40]. Oxidative damage has been proposed to be involved in the hypofunction of lacrimal glands [41]. One of the toxic mechanisms of PM exposure is oxidative stress, mediated by ROS [42]. In addition, the overproduction of ROS results in the increase of the apoptosis-related signaling pathways. HMGB1 acts as an extracellular signaling agent in cells damaged by oxidative stress and apoptosis [43,44]. Moreover, HMGB1 can translocate into the cytoplasm of cells with oxidative stress [45]. 8-OHdG is a biomarker of oxidative DNA damage [46,47]. In this study, the topical exposure of UPM on the ocular surface induced an increase in the apoptotic injury of lacrimal gland cells via oxidative stress through the increase in HMGB1 and 8-OHdG. Aucubin reduced this oxidative injury in lacrimal glands, which confirmed that aucubin has an antioxidative effect.

In the present study, we also showed that RAGE expression was highly increased in the lacrimal gland of UPM-exposed eyes. HMGB1 is a specific ligand for RAGE [23,25] Hori, O.; Brett, J.; Slattery, T.; Cao, R.; Zhang, J.; Chen, J.X.; Nagashima, M.; Lundh, E.R.; Vijay, S.; Nitecki, D.; Morser, J.; Stern, D.; Schmidt, A.M. *J. Biol. Chem.*
**1995**, *270*, 25752–25761. View Record in Scopus|Cited By in Scopus (441), whose expression was increased in the lacrimal glands of diabetic rats with tear film dysfunction [48]. The interaction of RAGE and HMGB1 can control oxidative stress and lead to proinflammatory effects [49]. Ethyl pyruvate, an inhibitor of HMGB1, reduced HMGB1 cytoplasmic translocation and RAGE expression along with ROS generation in microglial cells exposed to airborne organic dust [50]. Thus, the HMGB1/RAGE axis might induce oxidative injury in the lacrimal gland. In addition, the present study suggests that aucubin prevents UPM-induced oxidative injury in lacrimal glands by attenuating the HMGB1/RAGE signaling pathway.

## 4. Materials and Methods

### 4.1. Materials

Aucubin, with purity greater than 98%, was obtained from Sigma-Aldrich (Burlington, MA, USA). UPM was purchased from National Institute of Standards and Technology (SRM1648a, Gaithersburg, MD, USA). The phenol red cotton threads were supplied by FCI Ophthalmics (Zone Quick, Pembroke, MA, USA). The Rat ROS ELISA Kit and the in situ cell death detection kit were supplied by MyBioSource (San Diego, CA, USA) and Roche (Mannheim, Germany), respectively. Antibodies for 8-hydroxydeoxyguanosine (8-OhdG) and high mobility group box 1 (HMGB1) were purchased from Abcam (Boston, MA, USA). Antibody for advanced glycation end product (RAGE) was purchased from Santa Cruz Biotechnology (Paso Robles, CA, USA).

### 4.2. Animal Experiment

Six-week-old male SD rats (Orient Bio, Seongnam, Korea) were used to induce dry eye disease using UPM (National Institute of Standards and Technology, MD, USA) with a procedure described previously [51]. To induce dry eye disease, the eyes of rats in all groups except the control group were topically exposed to 20 µL of UPM (20 mg/mL) thrice a day for 5 d. Rats in the control group were exposed to same volume of PBS. After 5 d of UPM treatment, we selected the rats with a tear volume of less than two-thirds of the mean tear volume in the control rats and randomly assigned them to four groups: (1) normal control group, (2) UPM-treated group, (3) 0.1% aucubin-eye-drop-treated group (0.1% aucubin), (4) 0.5% aucubin-eye-drop-treated group (0.5% aucubin). Aucubin eye drops (0.1 and 0.5%) were prepared by dissolving aucubin in phosphate-buffered saline (PBS). The osmolality of aucubin eye drops was 290 mOsmol/kg. All eye drops were instilled at a dose of 20 μL per eye 30 min after exposure to 20 µL of UPM (20 mg/mL) thrice a day for an additional 5 d. At necropsy, lacrimal glands were isolated. Design of animal experiment and treatment are summarized in Figure 6. The experiment using animals conducted in this study was performed according to the protocol approved by Institutional Animal Care and Use Committee (IACUC approval No. 21-101).

### 4.3. Tear Secretion Analysis

The amount of tears was measured using the previously reported method [4]. All mice were anesthetized with ketamine (80 mg/kg) and xylazine (10 mg/kg). After placing phenol red cotton threads (Zone Quick; FCI Ophthalmics, Pembroke, MA, USA) in the outer cantus for one min, the length of the thread turned red by the tear fluid was measured. The amount of tears was expressed in millimeters.

### 4.4. Corneal Irregularity Score

Corneal irregularity induced by the tear film instability was evaluated using the previously reported method [4]. All mice were anesthetized with ketamine (80 mg/kg) and xylazine (10 mg/kg). A ring-shaped light from the fiber-optic ring illuminator of a stereomicroscope (SZ51; Olympus, Tokyo, Japan) was projected on the corneal surface, and the reflected lines of light were captured with a DP21 digital camera (Olympus, Tokyo, Japan). The line shape reflected on the surface of the cornea using a circular light of the ring illuminator attached to the stereomicroscope (Olympus, Tokyo, Japan) was scored according to the following criteria: 0, perfect circle shape; 1, distortion at 1/4 of the circle; 2, distortion at 2/4 of the circle; 3, distortion at 3/4 of the circle; 4, distortion across 4/4 of a circle; and 5, severe distortion that does not recognize the shape of the circle.

### 4.5. Oxidative Stress Assay

Frozen lacrimal glands were homogenized in lysis buffer (150 mM NaCl, 1% Triton X-100, and 10 mM Tris, pH 7.4). The homogenate was centrifuged at 10,000× *g* for 10 min at 4 °C, and the supernatant was collected for measurement of reactive oxygen species (ROS) levels. Total levels of reactive oxygen species (ROS) in lacrimal glands were analyzed using a Rat ROS ELISA Kit (MyBioSource, San Diego, CA, USA) in accordance with the manufacturer’s instructions.

### 4.6. TUNEL Staining

In the lacrimal glands, apoptosis was observed using an in situ cell death detection kit (Roche, Mannheim, Germany) in accordance with the manufacturer’s instructions. After paraffin removal, the cut sections were incubated in proteinase K (20 µg/mL) for 10 min at ambient temperature. Then, they were transferred into a TUNEL reaction mixture with a volume of 50 μL and placed in a dark incubator for 1 h at 37 °C. The number of TUNEL-positive apoptotic cells was calculated under a BX51 fluorescence microscope (Olympus, Tokyo, Japan).

### 4.7. Immunohistochemistry

Immunohistochemistry of 8-OhdG, HMGB1, and RAGE was performed using the previously reported method [4]. The lacrimal glands were removed from rats 10 d after UPM treatment. The formalin-fixed and paraffin-embedded tissue sections were labeled with anti-8-hydroxydeoxyguanosine (8-OHdG) antibody (Abcam, Boston, MA, USA), anti-high mobility group box 1 (HMGB1) antibody (Abcam, Boston, MA, USA), and anti-receptor for advanced glycation end product (RAGE) antibody (Santa Cruz, CA, USA). Then, sections were detected with a VECTASTAIN Elite ABC Universal Kit (Vector Laboratory, Burlingame, CA, USA). All sections were observed under a light microscope (Olympus, Tokyo, Japan). Signal intensity was measured using ImageJ software (NIH, Boston, MA, USA).

### 4.8. Quantitative RT-PCR

Total RNA was extracted from the frozen lacrimal glands using RNeasy Mini Kit (Qiagen, Hilden, Germany). Reverse transcription for cDNA synthesis was performed with 1 μg of each total RNA using SuperScript VILO Master Mix (Life Technologies, Carlsbad, CA, USA). Each cDNA was subjected to real-time quantitative polymerase chain reaction (qPCR) using TaqMan Master Mix together with TaqMan Gene Expression Assays for HMGB1 (Rn02377062_g1), RAGE (Rn01525753_g1), and β-actin (Rn01412977_g1) (Applied Biosystems, Bedford, MA, USA), performed on Step One Plus equipment (Thermo Fisher Scientific, Waltham, MA, USA). Standard thermocycling conditions were used in a total volume of 10 μL. Fold change in intensity of the target gene was calculated using Step One Software (Thermo Fisher Scientific, Waltham, MA, USA).

### 4.9. Statistical Analysis

Statistical analysis was performed using one-way analysis of variance (ANOVA) followed by Tukey’s multiple comparison test between groups using Prism 8.0 software (GraphPad, San Diego, CA, USA).

## 5. Conclusions

Our results showed that the topical administration of aucubin eye drops may provide a beneficial pharmacological option that prevents UPM-induced dry eye disease by increasing tear volume and inhibiting oxidative damage to the lacrimal glands.

## Figures and Tables

**Figure 1 molecules-27-02926-f001:**
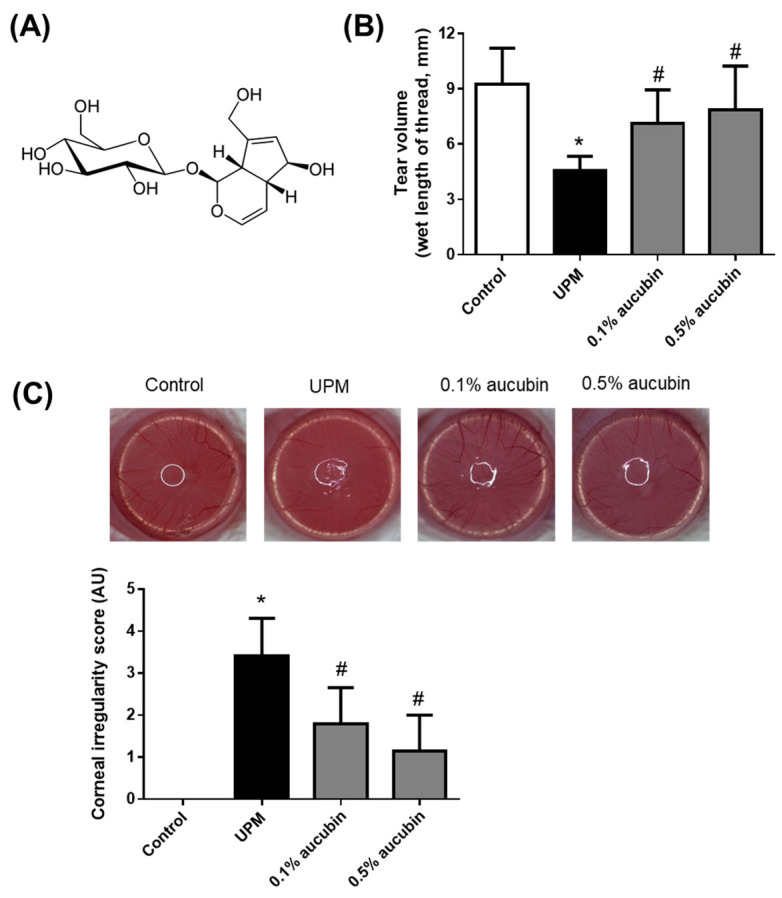
Effects of aucubin eye drops on tear hyposecretion, and corneal irregularity in UPM-induced dry eye rats. (**A**) The chemical structure of aucubin. (**B**) The tear volume was measured using a phenol red thread. (**C**) Representative reflected images of the ring illuminator on the corneal surface. Bar graph indicates the quantitative analysis of the corneal irregularity score. Data shown are mean ± standard deviation (SD) (*n* = 8). * *p* < 0.05 vs. Control group, # *p* < 0.05 vs. UPM group. Control group: Control, UPM-treated group: UPM, UPM + 0.1% aucubin-eye-drop-treated group: 0.1% aucubin, and UPM + 0.5% aucubin-eye-drop-treated group: 0.5% aucubin.

**Figure 2 molecules-27-02926-f002:**
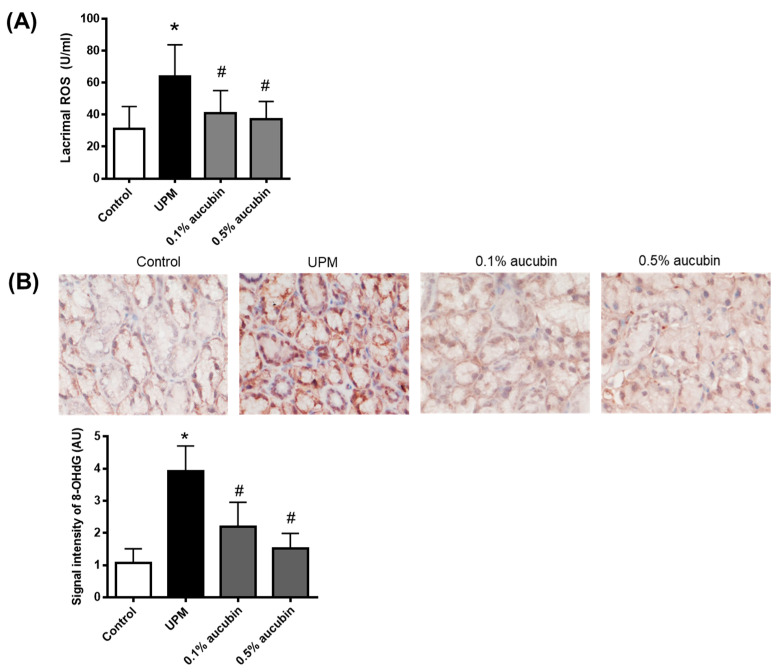
Effect of aucubin eye drops on ROS generation in lacrimal glands of UPM-induced dry eye rats. (**A**) Total levels of ROS in the lacrimal gland. (**B**) Immunohistochemical staining for 8-OHdG. Bar graph indicates the quantification of the 8-OHdG signal intensity. Data shown are mean ± SD (*n* = 8). * *p* < 0.05 vs. Control group, # *p* < 0.05 vs. UPM group. Control group: Control, UPM-treated group: UPM, UPM + 0.1% aucubin-eye-drop-treated group: 0.1% aucubin, and UPM + 0.5% aucubin-eye-drop-treated group: 0.5% aucubin.

**Figure 3 molecules-27-02926-f003:**
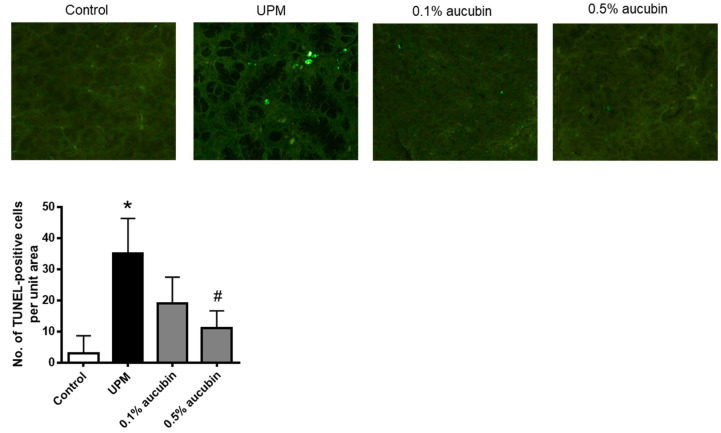
Effect of aucubin eye drops on apoptosis in lacrimal glands of UPM-induced dry eye rats. Representative images for TUNEL staining are presented. The numbers of TUNEL-positive cells were counted. Data shown are mean ± SD (*n* = 8). * *p* < 0.05 vs. Control group, # *p* < 0.05 vs. UPM group. Control group: Control, UPM-treated group: UPM, UPM + 0.1% aucubin-eye-drop-treated group: 0.1% aucubin, and UPM + 0.5% aucubin-eye-drop-treated group: 0.5% aucubin.

**Figure 4 molecules-27-02926-f004:**
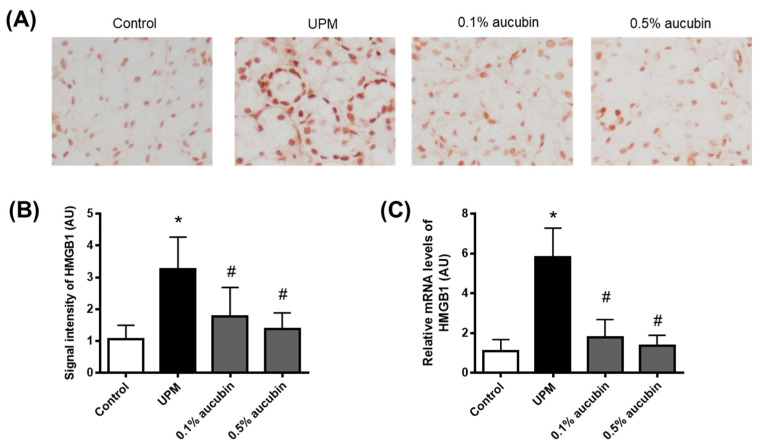
Effect of aucubin eye drops on the expression of HMGB1 in lacrimal glands of UPM-induced dry eye rats. (**A**) Representative images for immunohistochemical staining for HMGB1. (**B**) Bar graph indicates the quantification of the HMGB1 signal intensity. (**C**) Relative mRNA levels of HMGB1 were assessed using a quantitative RT-PCR. Data shown are mean ± SD (*n* = 8). * *p* < 0.05 vs. Control group, # *p* < 0.05 vs. UPM group. Control group: Control, UPM-treated group: UPM, UPM + 0.1% aucubin-eye-drop-treated group: 0.1% aucubin, and UPM + 0.5% aucubin-eye-drop-treated group: 0.5% aucubin.

**Figure 5 molecules-27-02926-f005:**
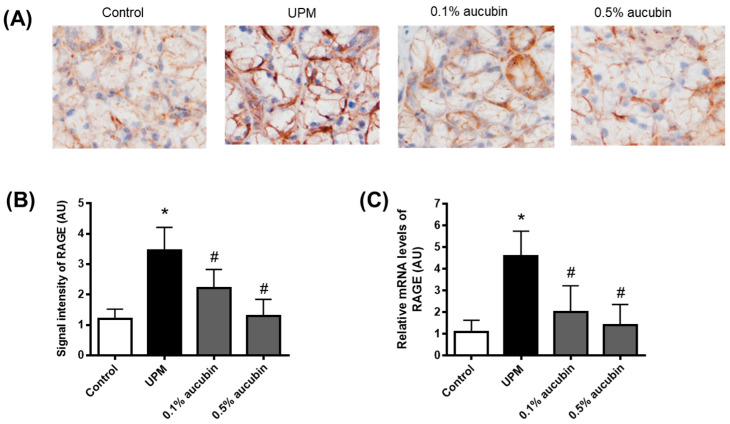
Effect of aucubin eye drops on the expression of RAGE in lacrimal glands of UPM-induced dry eye rats. (**A**) Representative images for immunohistochemical staining for RAGE. (**B**) Bar graph indicates the quantification of the RAGE signal intensity. (**C**) Relative mRNA levels of RAGE were assessed using a quantitative RT-PCR. Data shown are mean ± SD (*n* = 8). * *p* < 0.05 vs. Control group, # *p* < 0.05 vs. UPM group. Control group: Control, UPM-treated group: UPM, UPM + 0.1% aucubin-eye-drop-treated group: 0.1% aucubin, and UPM + 0.5% aucubin-eye-drop-treated group: 0.5% aucubin.

**Figure 6 molecules-27-02926-f006:**
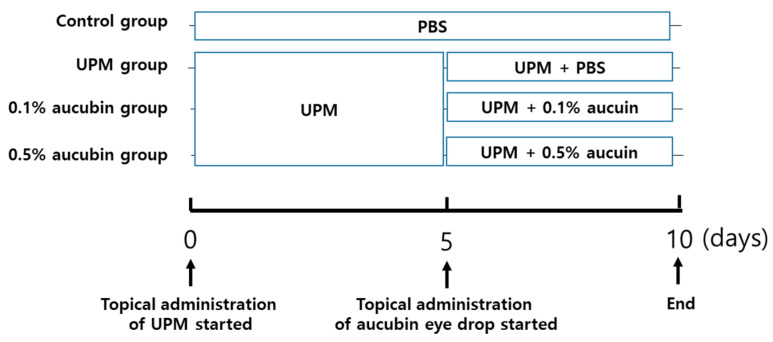
Experimental design for the animal study.

## Data Availability

The data presented in this study are available on request from the corresponding author.

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
