# Peer review of "Effect of Aucubin-Containing Eye Drops on Tear Hyposecretion and Lacrimal Gland Damage Induced by Urban Particulate Matter in Rats"

_molecules, 2022, doi:10.3390/molecules27092926_

Round 1

Reviewer 1 Report

Park et al. have studied the effect of aucubin-containing eye drop on tear hyposecretion and lacrimal gland damage induced by urban particulate matter. The work is technically good and has been done methodically, deserving a publication in molecules. However, the authors need to address the following issues before it could be accepted for publication:

  1. Abstract & Keywords – (1) “TUNEL staining” is missing. (2) Some optimized quantitative values should be given. (3) two more keywords such as ‘tear secretion’ and ‘corneal irregularity’ or some other related keywords should be included.
  2. Introduction – Chemical structure of Aucubin as well as some basic physico-chemical properties of Aucubin should be provided as one figure.
  3. Materials and methods – (1) “Materials” sub-section is missing. (2) It will be better be elaborate the methods for sub-section 4.2, 4.3, 4.4 and 4.5.
  4. Line 73 – give the full form of “SE” in first instance and abbreviate thereafter.
  5. Line 119-121 – why there is a reference Hori et al. [23] is provided in full form here, which is supposed to be only in “Reference” section.
  6. A schematic diagram summarizing the animal experiment design, parameters determined (tear secretion analysis, corneal irregularity, oxidative stress, tunel staining, immunohistochemistry etc.) and the trends observed on these parameters upon treatments in rats should be included.
  7. A conclusion section is missing!!
  8. Units – the units “month, day, minute, hour and second” should be abbreviated as “mo, d, min, h and s” throughout the manuscript.

Author Response

Abstract & Keywords – (1) “TUNEL staining” is missing. (2) Some optimized quantitative values should be given. (3) two more keywords such as ‘tear secretion’ and ‘corneal irregularity’ or some other related keywords should be included.

Answer: According to reviewer’s suggestion, I revised the Abstract section and added two keywords.

Introduction – Chemical structure of Aucubin as well as some basic physico-chemical properties of Aucubin should be provided as one figure.

Answer: The chemical structure of aucubin was added in Fig. 1A.

Materials and methods – (1) “Materials” sub-section is missing. (2) It will be better be elaborate the methods for sub-section 4.2, 4.3, 4.4 and 4.5.

Answer: According to reviewer’s suggestion, I revised the Materials and methods section.

Line 73 – give the full form of “SE” in first instance and abbreviate thereafter.

Answer: I revised it.

Line 119-121 – why there is a reference Hori et al. [23] is provided in full form here, which is supposed to be only in “Reference” section.

Answer: I apologize for the terrible typographical errors. I revised it.

A schematic diagram summarizing the animal experiment design, parameters determined (tear secretion analysis, corneal irregularity, oxidative stress, tunel staining, immunohistochemistry etc.) and the trends observed on these parameters upon treatments in rats should be included.

Answer: According to reviewer’s suggestion, the schematic diagram summarizing the animal experiment design was added in Fig. 6.

A conclusion section is missing!!

Answer: According to reviewer’s suggestion, the conclusion section was added

Units – the units “month, day, minute, hour and second” should be abbreviated as “mo, d, min, h and s” throughout the manuscript.

Answer: I revised it.

Reviewer 2 Report

I consider that experimental results can be improved as it follows:

- Line 16:

A dose-dependent effect must be statistically demonstrated. In fact, the Authors must demonstrate that the effect induced by a dose is significantly changed (increased or decreased) compared to the one caused of the next or previous dose.

-Line 104:

To show the effect caused by aucubin eye drops on the expression of HMGB1 in lacrimal glands, experiments of western blot and/or qPCR on tissue extracts should be added. In my opinion, measuring the immunohistochemical signal intensity is not sufficient.

-Line 118:

Similarly, in order to demonstrate the effect caused by aucubin on RAGE expression in lacrimal glands, experiments of western blot and/or qPCR on tissue extracts should be added. In my opinion, measuring the immunohistochemical signal intensity is not sufficient.

- Lines 119-121:

The reference should be moved to bibliographic paragraph.

Also, why is RAGE called a potential HMGB1 receptor?

- Statistical analysis can be improved.In graphs, the Authors should replace SE with Standard Deviation values in order to indicate the variation around the average. In addition, the Authors should report how many independent experiments were performed.

Author Response

- Line 16:

A dose-dependent effect must be statistically demonstrated. In fact, the Authors must demonstrate that the effect induced by a dose is significantly changed (increased or decreased) compared to the one caused of the next or previous dose.

Answer: I completely agree with the reviewer’s opinion. For a better understanding, I re-described this sentence.

-Line 104:

To show the effect caused by aucubin eye drops on the expression of HMGB1 in lacrimal glands, experiments of western blot and/or qPCR on tissue extracts should be added. In my opinion, measuring the immunohistochemical signal intensity is not sufficient.

Answer: According to reviewer’s suggestion, I performed the qPCR for HMGB1 in lacrimal glands. I added the result in Fig. 4.

-Line 118:

Similarly, in order to demonstrate the effect caused by aucubin on RAGE expression in lacrimal glands, experiments of western blot and/or qPCR on tissue extracts should be added. In my opinion, measuring the immunohistochemical signal intensity is not sufficient.

Answer: Answer: According to reviewer’s suggestion, I performed the qPCR for RAGE in lacrimal glands. I added the result in Fig. 5.

- Lines 119-121:

The reference should be moved to bibliographic paragraph.

Answer: I apologize for the terrible typographical errors. I revised it.

Also, why is RAGE called a potential HMGB1 receptor?

Answer: RAGE is a multiligand receptor that binds structurally diverse molecules, including HMGB1. For a better understanding, I re-described this sentence.

- Statistical analysis can be improved.In graphs, the Authors should replace SE with Standard Deviation values in order to indicate the variation around the average. In addition, the Authors should report how many independent experiments were performed.

Answer: According to reviewer’s suggestion, I revised it. All of the experimental data were taken from eight animals in each group (n=8).

Reviewer 3 Report

It would be convenient to include into the title,  the words “in rats”

Aucubin eye drop has an inhibitory activity against UPM-induced hypofunction of lacrimal glands by inhibiting ROS generation. The topical administration of aucubin eye drop may provide a beneficial pharmacological option that prevent UPM-induced dry eye disease

Topical administration of aucubin eye drops showed the beneficial effect against UPM-induced abnormal ocular changes,such as tear hyposecretion, and lacrimal gland damage. Therefore, our results reveal the pharmacological activities of aucubin in dry eye disease.

In conclusion, our results showed that the topical administration of aucubin eye drop may provide a beneficial pharmacological option that prevent UPM-induced dry eye disease by increasing tear volume, inhibiting oxidative damage to the lacrimal glands.

Author Response

According to reviewer’s suggestion, I revised the title. The entire manuscript was carefully proofread and corrected.

Round 2

Reviewer 2 Report

My suggestions have been applied. I think this version of the paper acceptable.